# Assessment of Tenderness and Anthocyanin Content in Zijuan Tea Fresh Leaves Using Near-Infrared Spectroscopy Fused with Visual Features

**DOI:** 10.3390/foods14172938

**Published:** 2025-08-22

**Authors:** Shuya Chen, Fushuang Dai, Mengqi Guo, Chunwang Dong

**Affiliations:** 1Tea Research Institute, Shandong Academy of Agricultural Sciences, Jinan 250033, China; 2Shandong Guohe Industrial Technology Institute Co., Ltd., Jinan 250014, China; 3College of Biosystems Engineering and Food Science, Zhejiang University, Hangzhou 310058, China

**Keywords:** near-infrared spectroscopy, Zijuan tea, anthocyanins, tenderness grading, visual features

## Abstract

Focusing on the characteristic tea resource Zijuan tea, this study addresses the difficulty of grading on production lines and the complexity of quality evaluation. On the basis of the fusion of near-infrared (NIR) spectroscopy and visual features, a novel method is proposed for classifying different tenderness levels and quantitatively assessing key anthocyanin components in Zijuan tea fresh leaves. First, NIR spectra and visual feature data were collected, and anthocyanin components were quantitatively analyzed using UHPLC-Q-Exactive/MS. Then, four preprocessing techniques and three wavelength selection methods were applied to both individual and fused datasets. Tenderness classification models were developed using Particle Swarm Optimization–Support Vector Machine (PSO-SVM), Random Forest (RF), and Convolutional Neural Networks (CNNs). Additionally, prediction models for key anthocyanin content were established using linear Partial Least Squares Regression (PLSR), nonlinear Support Vector Regression (SVR) and RF. The results revealed significant differences in NIR spectral characteristics across different tenderness levels. Model combinations such as TEX + Medfilt + RF and NIR + Medfilt + CNN achieved 100% accuracy in both training and testing sets, demonstrating robust classification performance. The optimal models for predicting key anthocyanin contents also exhibited excellent predictive accuracy, enabling the rapid and nondestructive detection of six major anthocyanin components. This study provides a reliable and efficient method for intelligent tenderness classification and the rapid, nondestructive detection of key anthocyanin compounds in Zijuan tea, holding promising potential for quality control and raw material grading in the specialty tea industry.

## 1. Introduction

The high-quality development of the tea industry relies on precise control over tea quality [1]. Zijuan tea, a unique tea cultivar, is distinguished by its tenderness level and intrinsic components—particularly anthocyanins—which are key determinants of tea quality. Due to its high anthocyanin content, Zijuan tea offers both distinctive health benefits and flavor, attracting strong consumer attention [2]. Chen et al., conducted a systematic analysis of characteristic metabolites in Zijuan tea from different regions in Yunnan Province, revealing significant differences in the content of tea polyphenols, amino acids, catechins, caffeine, sugars, and anthocyanins [3].

Traditional methods using visual and morphological cues such as leaf color and shape to determine the tenderness of fresh tea leaves rely heavily on manual expertise. These sensory-based approaches are highly subjective, inefficient, and poorly suited to the demands of large-scale production. However, anthocyanin content is typically measured using chemical techniques such as high-performance liquid chromatography (HPLC), which, while being highly accurate, are time-consuming, costly, complex, and inherently destructive, making them unsuitable for real-time online monitoring [4]. As a result, there is an urgent need within the tea industry for rapid, nondestructive, and accurate detection technologies to support digitalization and intelligent upgrading. Near-infrared spectroscopy (NIRS), which leverages the absorption characteristics of molecular bonds in the near-infrared region, enables the fast acquisition of multi-component information from samples; hence, it has been widely applied in the detection of agricultural product composition [5,6,7]. The advantages of NIRS lie in its nondestructive nature and high efficiency, enabling the rapid quantitative analysis of components such as anthocyanins. However, NIRS is susceptible to interference from the physical properties of samples, such as color and texture, and single-source spectral data often struggle to capture the complex characteristics of fresh leaves [8]. Nonetheless, visual features—such as color and texture—contain intuitive information related to leaf tenderness. Zijuan tea leaves of different tenderness levels exhibit noticeable color differences (due to variations in chlorophyll and anthocyanin accumulation) and changes in surface characteristics (such as leaf folding and vein density), reflecting the developmental stages. Integrating visual features with NIR spectroscopy allows for the fusion of macroscopic morphological and microscopic compositional information, compensating for the limitations of each individual technique and providing a more comprehensive basis for both tenderness classification and component detection in fresh tea leaves.

At present, the integration of multi-source information is gaining momentum in the field of agricultural product detection. However, research specifically focused on Zijuan tea fresh leaves remains limited. As shown in Table 1. Existing tea quality detection studies are primarily concentrated on conventional tea types such as green tea and black tea [9,10,11,12,13,14,15]. Due to its unique varietal characteristics—particularly its high anthocyanin content and purplish leaf coloration—Zijuan tea exhibits distinct visual features and spectral response patterns compared to ordinary tea leaves [16]. Therefore, targeted fusion-based modeling research for this tea variety is urgently needed. In addition, a key challenge lies in the efficient extraction of visual features, such as optimizing color space conversion and selecting informative texture descriptors, and in constructing suitable models that integrate spectral and visual data. Developing such fusion models to enable accurate detection of both leaf tenderness and anthocyanin content remains a critical task to be accomplished.

This study is based on the unique characteristics of the Zijuan tea variety and innovatively integrates near-infrared spectroscopy with visual features such as color and texture of fresh Zijuan tea leaves, aiming to establish models for tenderness classification and key anthocyanin content detection. Four preprocessing methods were applied to eliminate environmental noise and electronic signal interference in the NIR spectra, and three wavelength selection techniques were implemented to extract informative spectral bands and further optimize model performance. Finally, both linear and nonlinear machine learning models, along with a deep learning Convolutional Neural Network (CNN) model, were developed to achieve the accurate classification of leaf tenderness and prediction of key anthocyanin components. This research seeks to overcome the limitations of traditional detection methods by providing a technological foundation for the intelligent grading and rapid quality evaluation of Zijuan tea fresh leaves, promoting the transition of the tea industry toward greater precision and intelligence while also enriching the theoretical framework for multi-source data fusion in the detection of specialty tea resources.

## 2. Materials and Methods

### 2.1. Sample Preparation

The Zijuan fresh leaves used in this study were collected from the tea plantation of the Tea Research Institute, Chinese Academy of Agricultural Sciences (Hangzhou, Zhejiang, China). Based on different developmental stages, the fresh leaves were categorized and harvested as one bud with one leaf, one bud with two leaves, one bud with three leaves, one bud with four leaves, and fully mature leaves. A portion of the freshly harvested leaves was sealed in airtight bags and transported to the laboratory for near-infrared (NIR) spectral scanning, while another portion was rapidly frozen in liquid nitrogen (to prevent metabolite degradation) and transported on dry ice for anthocyanin content analysis. The brief flowchart of this study is shown in Figure 1.

### 2.2. Determination of Key Physicochemical Components in Zijuan Tea Fresh Leaves

#### 2.2.1. Anthocyanin Extraction

The collected Zijuan tea fresh leaves were completely freeze-dried using a vacuum freeze dryer. Before this operation, the sample chamber was installed, and the sealing gasket was checked to ensure it was properly in place without any air leaks. The machine was pre-cooled for 30 min with the refrigeration temperature set to −30 °C. Mesh bags containing samples of different tenderness levels, which had been labeled accordingly, were placed inside the sample chamber. After sealing, vacuum was applied until the pressure dropped below 50 mTorr. The samples were freeze-dried completely over a period of three days. Finally, the dried samples were removed and ground into powder.

Anthocyanins were extracted using an acidified ethanol ultrasonic-assisted method. Briefly, 0.5 g of Zijuan tea freeze-dried powder was weighed and mixed with 10 mL of 1% (*w*/*v*) citric acid–60% (*v*/*v*) ethanol solution (pH 2.0). The mixture was vortexed for 1 min to ensure thorough wetting, and then ultrasonic extraction was performed in an ice bath protected from light at 200 W for 10 min while maintaining the temperature below 25 °C. The extract was subsequently centrifuged at 8000 rpm and 4 °C for 10 min. The supernatant was transferred to a 25 mL brown volumetric flask. The residue was washed with 5 mL of the same acidified ethanol solution and centrifuged again, and then the supernatants were combined and brought to volume with the extraction solvent to 25 mL. The solution was filtered through a 0.45 μm nylon membrane filter, and the filtrate was immediately used for analysis.

#### 2.2.2. UHPLC-Q-Exactive/MS Analysis

Anthocyanin analysis was performed using the UHPLC-Q-Exactive/MS system. The chromatographic conditions were as follows: an Accucore™ C18 column (2.1 × 100 mm, 2.6 μm) was used. A binary gradient elution was applied at a flow rate of 0.3 mL/min. The mobile phases consisted of 0.1% formic acid in water (A) and 0.1% formic acid in acetonitrile (B). The gradient program was set as follows: 0 min, 5% B; 0–10 min, increased to 20% B; 10–15 min, increased to 30% B; 15–18 min, increased to 95% B and held for 2 min; 20 min, returned to 5% B and held for 2 min. The column temperature was maintained at 35 °C, and the injection volume was 2 μL.

The mass spectrometry conditions were set as follows: HESI-II heated electrospray ionization source; positive ion mode; resolution of 70,000 (Full MS) and 17,500 (dd-MS2); spray voltage of 3.8 kV; capillary temperature at 320 °C; auxiliary gas temperature set to 350 °C with a flow rate of 10 L/min; scan range (*m*/*z*) from 150 to 1500.

#### 2.2.3. Standard Solution Preparation

Standard compounds including cya-3-O-rutinoside, cya-3-O-galactoside, cya-3-O-glucoside, peonidin-3-O-galactoside, peonidin, peonidin-3-O-glucoside, malvin, delphinidin-3-O-galactoside, delphinidin-3-O-glucoside, pelargonidin-3-O-galactoside, pelargonidin-3-O-glucoside, pelargonidin, cyanidin, delphinidin, petunidin, petunidin-3-O-glucoside, and malvin-3-O-glucoside with a purity of no less than 98% were purchased from Sigma. The standards were dissolved in a solvent consisting of 2% formic acid in methanol–water solution (*v*:*v*, 50:50) to prepare stock solutions at a concentration of 1 mg/mL. Subsequently, volumes of 0.5, 1.0, 2.5, 10.0, 20.0, or 50.0 μL were pipetted to prepare mixed standard solutions of varying concentrations. The stock solutions were stored at −20 °C. Data acquisition was performed on the standard solutions using the above-described mass spectrometry method to establish an anthocyanin standard database. Calibration curves were generated for quantitative analysis of anthocyanins in samples.

#### 2.2.4. Quantification of Anthocyanins in Zijuan Tea Fresh Leaves

The contents of major anthocyanins in tea leaf samples were calculated on the basis of calibration curves constructed from standard compounds.

### 2.3. Near-Infrared Spectral Acquisition and Spectral Preprocessing

Near-infrared spectral data were collected using an IAS3100 near-infrared spectrometer from Wuxi Intelligent Analysis Service Co., Ltd., Wuxi, China. Diffuse reflectance spectral scanning was performed on freshly harvested Zijuan tea leaves at room temperature. The laboratory relative humidity was maintained at approximately 60%, and the temperature was controlled around 20 °C. Tea samples were thoroughly compacted in Petri dishes to ensure uniformity. After preheating the instrument for 30 min, spectral data were acquired. Each sample was scanned 10 times, with the sample thoroughly mixed before each scan. The average of the 10 scans was taken as the final spectral data for a specific sample. In total, 100 spectral datasets were collected for building classification and prediction models. The spectral detection range was 900–1700 nm, with 801 wavelength points.

Near-infrared spectral data often contain interference signals such as baseline drift and low-frequency noise caused by environmental lighting and instrument dark current. In this study, four preprocessing algorithms were applied: Savitzky-Golay (S-G) first derivative filtering was used to eliminate baseline drift and enhance peak boundary recognition, with a window width of 11 and a polynomial order of 3 [17,18]; Standard Normal Variate (SNV) transformation was performed to remove baseline drift and spectral intensity variations through centering and scaling [19,20]; one-dimensional median filtering (Medfilt), a nonlinear signal processing method, was used to replace original spectral values with local median values within a sliding window to eliminate noise caused by instrument fluctuations and outliers from sudden environmental changes [21,22]; and normalization (Normaliz) was utilized to scale each spectrum to unit length by dividing by its Euclidean norm to eliminate variations due to differences in optical path length [23].

### 2.4. Extraction of Visual Color and Texture Features

A small image processing program developed using the GUI module of MATLAB software was used to extract color and texture features of Zijuan tea fresh leaves at different tenderness levels [24]. The color features included R, G, B, H, S, V, 2G-R-B, R/G, and hab*, while the texture features comprised smoothness (r), standard deviation (δ), entropy (e), uniformity (U), and mean gray level (m).

### 2.5. Data Fusion Strategies

Three data analysis methods were proposed to provide effective quality assessment and intuitive data support for Zijuan tea fresh leaves: (1) a discrimination model and anthocyanin quantitative detection model based on near-infrared spectroscopy; (2) a discrimination model and anthocyanin quantitative detection model based on color and texture features; and (3) a discrimination model and anthocyanin quantitative detection model based on data fusion. The three data fusion strategies are shown in Table 2. As shown in Figure 1, the data fusion method directly concatenates near-infrared spectral information with color and texture features. The fused data is then subjected to preprocessing and wavelength selection, followed by dimensionality reduction using PCA to establish models for determining the tenderness of Zijuan tea fresh leaves and predicting the content of key anthocyanins.

### 2.6. Establishment of Tenderness Classification Models for Zijuan Tea Fresh Leaves

In this study, tenderness classification models for Zijuan tea fresh leaves were developed using machine learning methods including Particle Swarm Optimization–Support Vector Machine (PSO-SVM) [25], Random Forest (RF) [26], and deep learning Convolutional Neural Network (CNN) [27]. PSO-SVM combines the ability of SVM to handle high-dimensional and small samples with the powerful automatic parameter optimization ability of PSO and can efficiently construct tenderness classification models with strong discriminative ability and good generalization performance. The high precision, robustness, automatic feature importance assessment capability, and relatively low parameter adjustment requirements of RF make it a powerful tool for building discriminative models. CNN is particularly suitable for discriminative tasks. It can automatically learn spectral features that the human eye may find difficult to describe or quantify, as well as subtle visual features related to tenderness. Prior to modeling, the data were preprocessed and subjected to Principal Component Analysis (PCA) [28]. The model input comprised the number of principal components (PCs) corresponding to the minimum root mean square error in the training set. The Kennard-Stone (K-S) algorithm [29] was used to divide the dataset into training and prediction sets at a ratio of 4:1, resulting in 80 samples for training and 20 samples for prediction.

### 2.7. Development of Prediction Models for Key Anthocyanin Content in Zijuan Tea Fresh Leaves

Despite the preprocessing steps, the near-infrared spectral data of Zijuan tea fresh leaves still contained a large amount of redundant information, which would lead to model overfitting or underfitting. To address this issue, this study employed three feature wavelength selection algorithms to significantly reduce the number of spectral wavelengths: Competitive Adaptive Reweighted Sampling (CARS) [30], Bootstrapping Soft Shrinkage (BOSS) [31], and Successive Projections Algorithm (SPA) [32].

After feature wavelength selection, Principal Component Analysis (PCA) was applied to the spectral data for dimensionality reduction. This method transforms the selected feature wavelengths into a smaller number of principal components, retaining the main spectral information while reducing the model input size and improving detection efficiency.

Prediction models for key anthocyanin content in Zijuan tea fresh leaves were developed using three traditional machine learning algorithms: Partial Least Squares Regression (PLSR) [20], nonlinear Support Vector Regression (SVR) [33], and Random Forest (RF) [34]. PLSR effectively handles spectral collinearity, avoiding the ill-conditioned solutions inherent in traditional linear regression. SVR accurately captures the nonlinear response between anthocyanins and spectral data by mapping the data to a high-dimensional space using the RBF or polynomial kernel function. RF demonstrates exceptional robustness to noisy spectral data (e.g., scattering interference) by reducing variance through bootstrap aggregation of hundreds of decision trees. Five-fold cross-validation was performed on the training set to optimize the number of decision trees for RF, the penalty parameter for SVR, and the optimal number of principal components for PLSR. Model performance was evaluated using the calibration set correlation coefficient (Rc), prediction set correlation coefficient (Rp), root mean square error of calibration (RMSEC), root mean square error of prediction (RMSEP), and relative percentage deviation (RPD) as the final metrics.

### 2.8. Data Analysis Software

Normality tests, homogeneity of variance tests and nonparametric tests in this study were performed using IBM SPSS Statistics 27 software. Discrimination and prediction models were developed using MATLAB 2019a. All plots were created using Origin 2022 software.

## 3. Results and Analysis

### 3.1. Analysis of Anthocyanin Content Differences in Zijuan Tea Fresh Leaves

In this study, a total of 16 anthocyanins were measured in Zijuan tea fresh leaves. Through statistical analysis, six anthocyanins with relatively high contents were classified as key anthocyanin components, while ten with lower contents were classified as trace anthocyanin components. The Kolmogorov-Smirnov (K-S) test was performed on the anthocyanin components at different tenderness levels, revealing that none followed a normal distribution. Homogeneity of variance tests showed significance levels below 0.05, indicating unequal variances. Therefore, the Kruskal–Wallis test for independent samples was used to analyze differences among tenderness levels. The differences in anthocyanin content across tenderness levels are shown in Figure 2, with tenderness grades defined from one bud one leaf to fully mature leaf as Level 1 through Level 5, respectively. As shown in Table 3, the total anthocyanin content peaked at Level 4, while fully mature leaves (Level 5) had the lowest anthocyanin content. Figure 2a indicates that only a few adjacent tenderness levels showed no significant differences (*p* > 0.05) in component content, while most key anthocyanin components exhibited significant or highly significant differences (*p* < 0.05) across tenderness levels. The key anthocyanin components—delphinidin-3-O-galactoside, delphinidin-3-O-glucoside, cyanidin-3,5-O-diglucoside, cyanidin-3-O-glucoside, petunidin, and cyanidin—had content ranges of 0.895–1.651 mg/g, 0.513–0.910 mg/g, 2.058–2.695 mg/g, 5.009–6.116 mg/g, 5.768–8.923 mg/g, and 0.179–0.789 mg/g, respectively. Figure 2b presents the distribution of trace anthocyanin components in Zijuan tea at different tenderness levels. The detailed content of trace anthocyanins is shown in Appendix A. The contents of cyanidin-3-O-rutinoside and pelargonidin-3,5-O-diglucoside generally increased with tenderness level. The contents of peonidin-3-O-galactoside, peonidin-3-O-glucoside, malvidin-3-O-glucoside, delphinidin, pelargonidin, peonidin, and malvidin first decreased and then increased as the tenderness level increased, reaching their highest values at Level 5. This indicates that trace anthocyanin components constitute the largest proportion in fully mature leaves.

### 3.2. Analysis of Visual Color and Texture Features

The Kruskal–Wallis test for independent samples was used to analyze differences in color and texture features among different tenderness levels, and violin plots were drawn. As depicted in Figure 3, there were generally no significant differences (*p* > 0.05) in color and texture features from Level 1 to Level 4, while Level 5 showed highly significant differences (*p* < 0.001) compared to Levels 1–4. As tenderness increased from Level 1 to Level 5, leaf maturity progressed. Anthocyanins, the core source of Zijuan tea’s purple color, are present in high concentrations that give the leaves a rich and uniform purple hue. The highest anthocyanin content was observed at Level 4, where leaf color transitioned from light purple to deep purple and then to green. At Level 5, the R, G, B, S, L, 2G-R-B, and mean gray level (m) were at their highest values, whereas R/G, standard deviation (δ), hue (H), and smoothness (r) were at their lowest.

### 3.3. PCA Cluster Analysis

To visually demonstrate the differences in anthocyanin content among different tenderness levels of Zijuan tea fresh leaves, PCA cluster analysis was performed on key anthocyanin components (Figure 4c) and trace components (Figure 4d). Clear clustering trends were observed among different tenderness levels, as shown in Figure 4c,d, indicating significant differences in both key and trace anthocyanin components. For the key anthocyanins, the first three principal components accounted for 62.9% (PC1), 20.7% (PC2), and 12.0% (PC3) of the variance, totaling 95.6% of the overall information. For the trace anthocyanins, PC1, PC2, and PC3 accounted for 77.3%, 16.7%, and 4.7% of the variance, respectively, covering 98.7% of the overall information. The 3D PCA clustering effectively captured the comprehensive information of the samples.

Figure 4a and Figure 4b show the PCA clustering of near-infrared spectra and color–texture features, respectively, for Zijuan tea fresh leaves at different tenderness levels. As seen in Figure 4a, Level 5 samples are distinctly clustered together, while Levels 1 to 4 are difficult to separate. Figure 4b reveals that PCA clustering based on color and texture features alone is insufficient to discriminate between different tenderness levels of Zijuan tea. Therefore, this study aimed to develop machine learning models to accurately classify the tenderness levels of Zijuan tea fresh leaves, providing a basis for selecting high-quality tea raw materials.

### 3.4. Preprocessing Near-Infrared Spectra of Zijuan Tea Fresh Leaves and Establishment of Tenderness Classification Models

The raw spectra are shown in Figure 5a. Due to significant noise in the 1650–1700 nm range, this segment was removed, retaining only the 900–1650 nm range and yielding 751 spectral features. The spectra after preprocessing by four methods—Savitzky-Golay (S-G), Standard Normal Variate (SNV), median filtering (Medfilt), and normalization (Normaliz)—are shown in Figure 5b–e. Figure 5f displays the average spectra of Zijuan tea fresh leaves at five tenderness levels. It can be observed that the absorbance decreases progressively from Level 1 to Level 5. As the leaf position moves lower on the plant, the leaves become more mature and darker in color; the high pigment content and dense structure inhibit diffuse reflectance. The more difficult the diffuse reflectance is, the lower the absorbance. The near-infrared spectra of Zijuan tea fresh leaves reflect their rich internal chemical composition. The absorption peak near 970–990 nm is related to the first overtone of O–H bonds in the molecules, mainly originating from moisture and free or weakly hydrogen-bonded hydroxyl groups (such as polyphenols and free water). Meanwhile, the absorption around 1150 nm is primarily associated with the first overtone of C–H bonds and combination vibrations of C–H bending. A strong absorption peak appears near 1390–1420 nm, related to the second overtone of O–H groups, which is one of the strongest water absorption peaks in the NIR region. The 1500–1600 nm region corresponds to the second overtone of C–H and combination bands of O–H, mainly characterizing complex organic compounds.

Tenderness classification models for Zijuan tea fresh leaves were developed using the machine learning methods PSO-SVM, Random Forest (RF) and deep learning CNN. Table 4, Table 5 and Table 6 present the accuracy of these classification models. Interestingly, among all models, Medfilt preprocessing consistently demonstrated excellent classification performance. For near-infrared spectroscopy (NIR), the PSO-SVM model with Medfilt preprocessing used 16 principal components (PCs) and achieved calibration and prediction accuracies of 96.50% and 90%, respectively; the RF model with 11 PCs achieved 100% and 95% accuracy on calibration and prediction sets, respectively; while the CNN model with 11 PCs achieved 100% accuracy on both training and prediction sets. For color and texture features (TEX), the PSO-SVM model with Medfilt preprocessing (10 PCs) reached 100% and 90% calibration and prediction accuracy; the RF model (10 PCs) achieved 100% accuracy on both sets; whereas the CNN model (10 PCs) obtained 97.5% calibration accuracy and 85% prediction accuracy. For the fused data (NIR + TEX), the PSO-SVM model with Medfilt preprocessing (9 PCs) achieved 100% calibration accuracy and 85% prediction accuracy; the RF and CNN models (25 PCs) both reached 100% calibration accuracy and 95% prediction accuracy. In summary, the TEX + Medfilt + RF and NIR + Medfilt + CNN combinations were the best-performing models for tenderness classification of Zijuan tea fresh leaves. The confusion matrix is shown in Figure 6. The data fusion of NIR and TEX did not significantly improve model performance, while it increased the input dimensionality.

### 3.5. Analysis of Near-Infrared Spectra of Zijuan Tea Fresh Leaves and Optimization of Data Fusion Preprocessing

Although the spectral trends among different samples are similar, there are subtle differences in absorbance that are difficult to detect with the naked eye. Therefore, a nonlinear Support Vector Regression (SVR) algorithm was employed to develop prediction models for key anthocyanin components. The number of spectral bands remained unchanged after preprocessing; however, to reduce model input dimensionality, Principal Component Analysis (PCA) was applied to the preprocessed spectra before modeling. The optimal number of principal components (PCs) was selected based on the smallest root mean square error of calibration (RMSEC) in the training set, and these PCs were used as input features for the model.

The evaluation metrics of prediction models for six key anthocyanin components and total anthocyanins, established using three methods—near-infrared spectroscopy (NIR), color–texture features (TEX), and fused NIR + TEX data—are presented in Table 7, Table 8, Table 9, Table 10, Table 11, Table 12 and Table 13. For the delphinidin-3-O-galactoside prediction model, the best preprocessing method for all three data types turned out to be Medfilt. Both NIR + Medfilt and NIR + TEX + Medfilt showed high accuracy but exhibited overfitting; therefore, the optimal model combination was TEX + Medfilt, which used the smallest number of principal components (PCs) and achieved Rc = 0.98, Rp = 0.98, and RPD = 4.687. In the delphinidin-3-O-glucoside model, SNV, S-G, and Medfilt were the best preprocessing methods for NIR, TEX, and NIR + TEX, respectively. Since NIR + SNV and NIR + TEX + Medfilt showed overfitting, the optimal preprocessing combination was TEX + S-G with only nine PCs, achieving Rc = 0.94, Rp = 0.96, and RPD = 3.718. For the cyanidin-3,5-O-diglucoside model, fused data showed better performance, with NIR + TEX + Medfilt as the optimal preprocessing combination, achieving Rc = 0.92, Rp = 0.93, and RPD = 2.736. In the cyanidin-3-O-glucoside model, the best prediction performance was obtained using Medfilt preprocessing on single NIR data, with Rc = 0.95, Rp = 0.95, and RPD = 3.267; other models exhibited overfitting. For the petunidin model, all preprocessed models showed overfitting, and the best combination was NIR + Medfilt. Further feature band selection will be applied in the future to improve prediction performance. In the cyanidin model, TEX + Normaliz showed underfitting, and the best prediction combination was NIR + Medfilt, with Rc = 0.93, Rp = 0.90, and RPD = 2.303. For the total anthocyanins model, TEX + S-G was the optimal combination, achieving Rc = 0.97, Rp = 0.96, and RPD = 3.280.

### 3.6. Near-Infrared Spectral Feature Band Selection

After data fusion and preprocessing optimization, the prediction accuracy of the best model combination for petunidin was unsatisfactory. Therefore, three feature band selection algorithms—CARS, BOSS, and SPA—were applied to further improve the model’s predictive performance (Table 14). As shown in Table 13, the CARS algorithm selected 52 feature bands from the 751 near-infrared spectral bands. After PCA dimensionality reduction, only 11 principal components were retained. The resulting model improved the accuracy of both the training and prediction sets by 0.02, with the RPD value increasing from 2.465 to 2.584, indicating only a slight improvement in model accuracy. The BOSS algorithm selected 20 feature bands, and after PCA, the input features were reduced to eight. While the training set accuracy increased by 0.02 compared to the original model, the prediction set accuracy decreased by 0.02, exacerbating overfitting. The RPD value also dropped significantly. The SPA algorithm selected 37 feature bands (Figure 7), which were reduced to five principal components by PCA. The model’s training set accuracy remained unchanged, but the prediction set accuracy increased by 0.04, significantly alleviating overfitting. The RPD value rose from 2.465 to 2.888, indicating stronger predictive performance. Therefore, for the key anthocyanin petunidin, the optimal combination was NIR + Medfilt + SPA, achieving Rc = 0.97, Rp = 0.95, and an RPD value of 2.888.

### 3.7. Optimization of Prediction Models for Key Anthocyanin Components

The nonlinear Support Vector Regression (SVR) model can effectively fit complex nonlinear relationships by using the kernel trick to find a function such that most data points lie within an ε-insensitive margin around it while minimizing the loss caused by points outside this margin (support vectors) [35]. Given the complex linear and nonlinear relationships in near-infrared spectra, this study also established linear Partial Least Squares Regression (PLSR) and nonlinear Random Forest (RF) models to comprehensively optimize the prediction of key anthocyanin components in Zijuan tea fresh leaves (Table 15). The PLSR models for total anthocyanins and key components showed signs of underfitting, whereas the RF models improved training accuracy compared to SVR but decreased prediction accuracy, intensifying model overfitting. Figure 8 shows scatter plots of actual versus predicted values for key anthocyanin components. According to model performance metrics, the best models were TEX + Medfilt + SVR for delphinidin-3-O-galactoside (Rc = 0.98, Rp = 0.98, RPD = 4.687); TEX + S-G + SVR for delphinidin-3-O-glucoside (Rc = 0.94, Rp = 0.96, RPD = 3.718) and cyanidin-3,5-O-diglucoside (Rc = 0.96, Rp = 0.94, RPD = 2.755); NIR + Medfilt + SVR for cyanidin-3-O-glucoside (Rc = 0.95, Rp = 0.95, RPD = 3.267) and cyanidin (Rc = 0.93, Rp = 0.90, RPD = 2.304); NIR + Medfilt + SPA + SVR for petunidin (Rc = 0.97, Rp = 0.95, RPD = 2.888); and TEX + S-G + SVR for total anthocyanins (Rc = 0.97, Rp = 0.96, RPD = 3.280).

## 4. Conclusions

This study employed near-infrared spectroscopy combined with visual features, integrating spectral preprocessing and variable selection techniques to develop models for Zijuan tea fresh leaf tenderness classification and key anthocyanin content prediction. From the results, the main conclusions can be drawn as follows:

The differences in anthocyanin content among different tenderness levels of Zijuan tea fresh leaves were systematically analyzed, finding significant or highly significant variation in key anthocyanin components across tenderness grades. Color and texture features showed no significant differences from one bud one leaf to one bud four leaves, while fully mature leaves differed highly significantly from these younger stages. PCA clustering effectively distinguished anthocyanin content among different tenderness levels. The best tenderness classification model combinations were found to be TEX + Medfilt + RF and NIR + Medfilt + CNN, both achieving 100% accuracy on training and prediction sets. Prediction models for key anthocyanin components were established, enabling quantitative prediction of six major anthocyanins and total anthocyanins. The optimal models were delphinidin-3-O-galactoside with TEX + Medfilt + SVR (Rc = 0.98, Rp = 0.98, RPD = 4.687); delphinidin-3-O-glucoside with TEX + S-G + SVR (Rc = 0.94, Rp = 0.96, RPD = 3.718); cyanidin-3,5-O-diglucoside with TEX + S-G + SVR (Rc = 0.96, Rp = 0.94, RPD = 2.755); cyanidin-3-O-glucoside with NIR + Medfilt + SVR (Rc = 0.95, Rp = 0.95, RPD = 3.267); petunidin with NIR + Medfilt + SPA + SVR (Rc = 0.97, Rp = 0.95, RPD = 2.888); cyanidin with NIR + Medfilt + SVR (Rc = 0.93, Rp = 0.90, RPD = 2.304); and total anthocyanins with TEX + S-G + SVR (Rc = 0.97, Rp = 0.96, RPD = 3.280). All models demonstrated excellent predictive performance. This research provides an important theoretical foundation for mechanized harvesting, intelligent grading, and quality evaluation of the specialty Zijuan tea resource, laying the groundwork for the digitalization and intelligent upgrading of the specialty tea industry.

## Figures and Tables

**Figure 1 foods-14-02938-f001:**
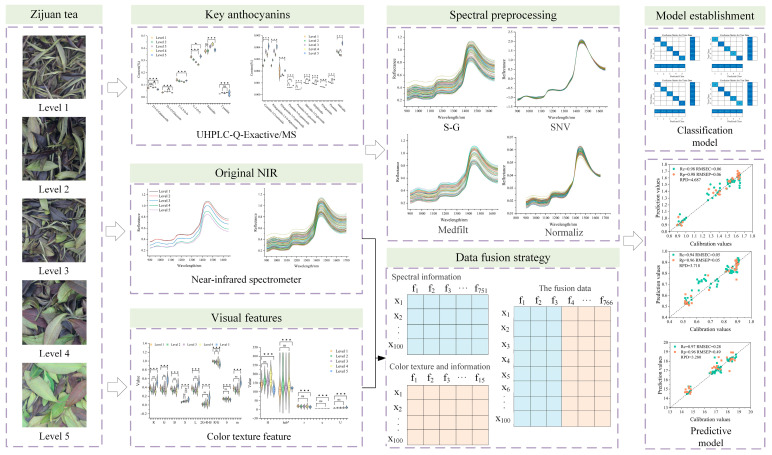
Research methodology flowchart (*p* value: ns *p* > 0.05; * *p* < 0.05; *** *p* < 0.001).

**Figure 2 foods-14-02938-f002:**
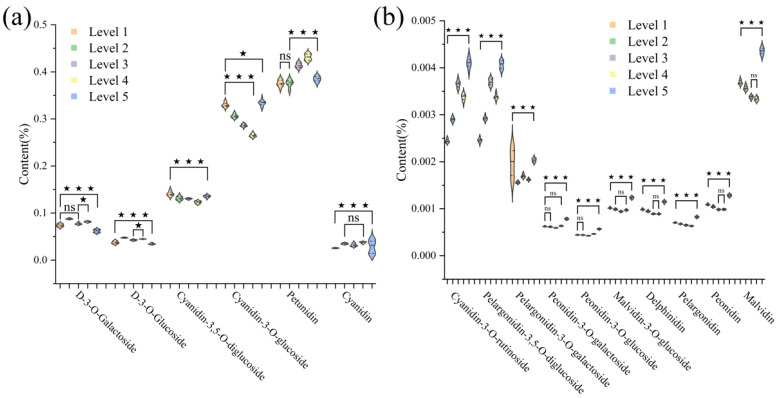
Content of anthocyanin components in Zijuan tea fresh leaves of varying tenderness levels: (**a**) six key anthocyanin components; (**b**) ten minor anthocyanin components (*p* value: ns *p* > 0.05; * *p* < 0.05; ** *p* < 0.01; *** *p* < 0.001).

**Figure 3 foods-14-02938-f003:**
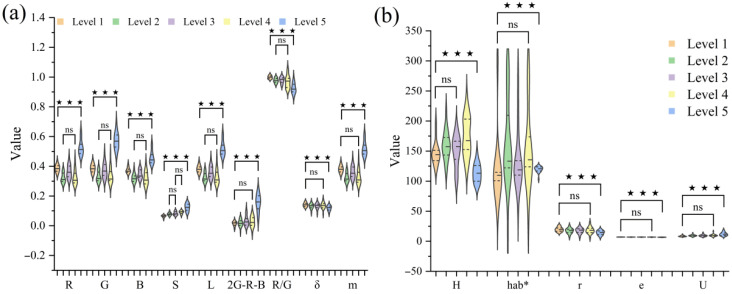
Violin plots of color and texture features in Zijuan tea fresh leaves across tenderness levels: (**a**) is the color feature and correlation test; (**b**) is the texture feature and correlation test (*p* value: ns *p* > 0.05; * *p* < 0.05; ** *p* < 0.01; *** *p* < 0.001).

**Figure 4 foods-14-02938-f004:**
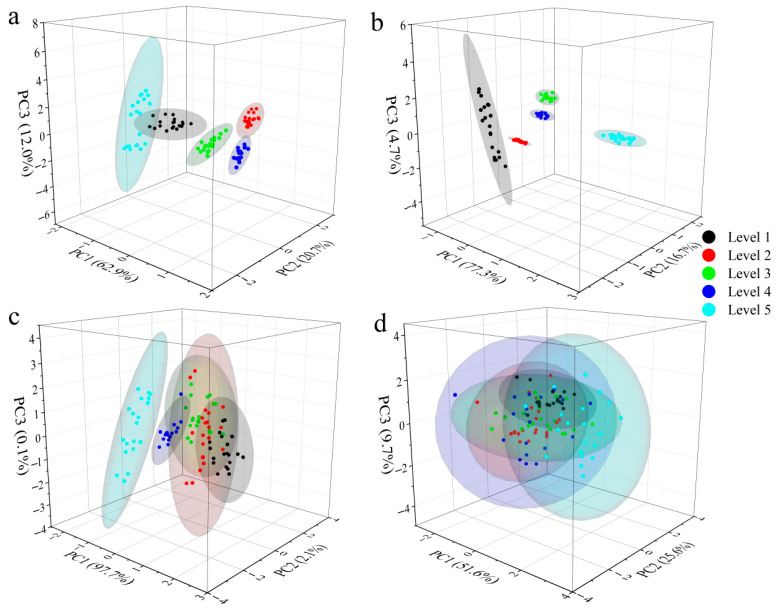
PCA of Zijuan tea fresh leaves across tenderness levels. (**a**) NIR spectra PCA clustering; (**b**) color–texture feature PCA; (**c**) key anthocyanin component PCA; (**d**) minor anthocyanin component PCA.

**Figure 5 foods-14-02938-f005:**
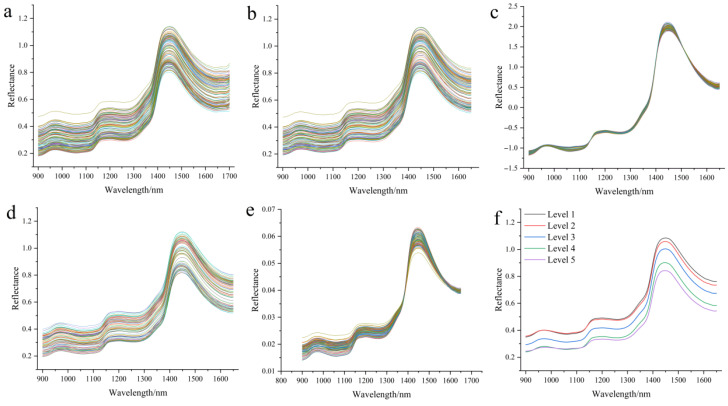
NIR of Zijuan tea fresh leaves: (**a**) raw spectra; (**b**) S-G preprocessed spectra; (**c**) SNV preprocessed spectra; (**d**) Medfilt preprocessed spectra; (**e**) Normaliz preprocessed spectra; (**f**) group mean spectra across tenderness levels.

**Figure 6 foods-14-02938-f006:**
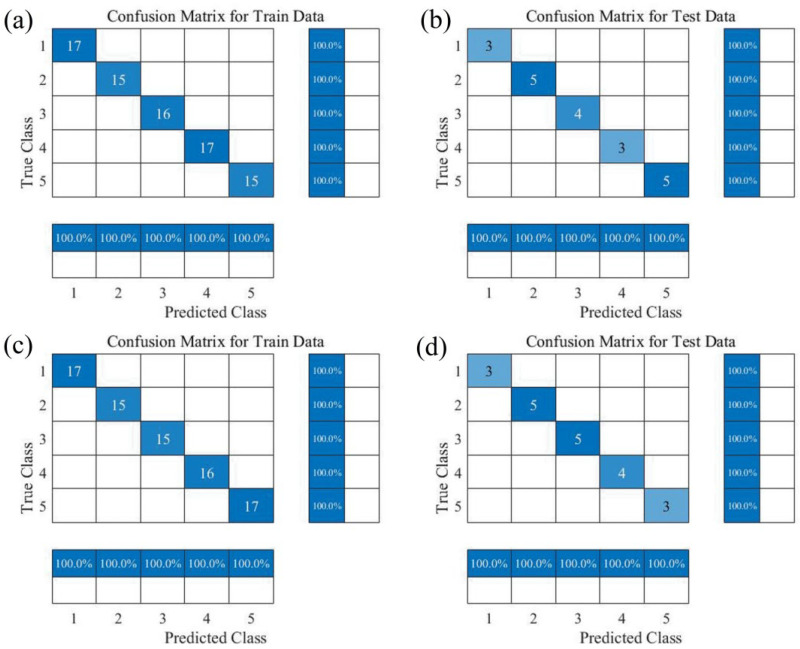
Confusion matrix diagrams for the optimal Zijuan tea tenderness discrimination model. (**a**) Training set of TEX + Medfilt + RF; (**b**) prediction set of TEX + Medfilt + RF; (**c**) training set of NIR + Medfilt + CNN; (**d**) prediction set of NIR + Medfilt + CNN.

**Figure 7 foods-14-02938-f007:**
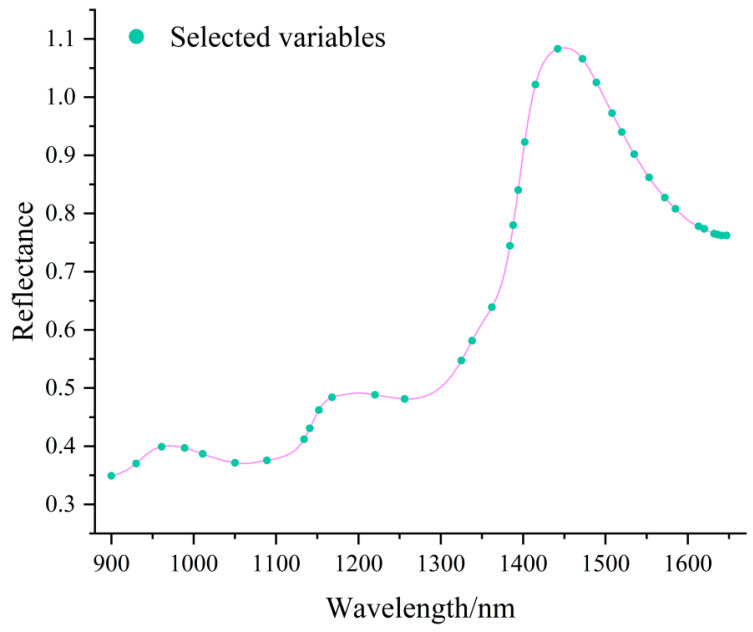
Waveband selection plot for NIR using SPA.

**Figure 8 foods-14-02938-f008:**
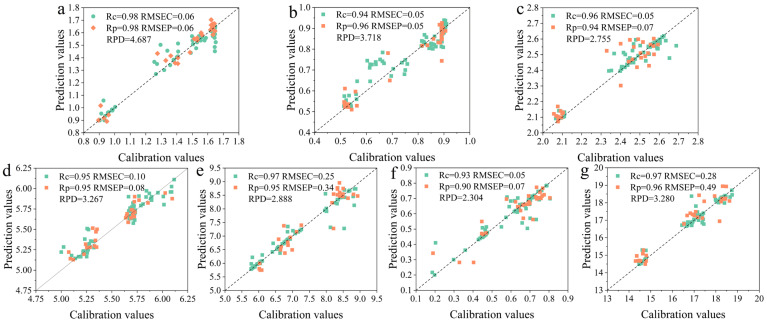
Scatter plot of prediction model for anthocyanins and key components of anthocyanins. The prediction of (**a**) delphinidin-3-O-galactoside by TEX + Medfilt + SVR, (**b**) delphinidin-3-O-glucoside by TEX + S-G + SVR, (**c**) cyanidin-3,5-O-diglucoside by TEX + S-G + SVR, (**d**) cyanidin-3-O-glucoside by NIR + Medfilt + SVR, (**e**) petunidin by NIR + Medfilt + SPA + SVR, (**f**) cyanidin by NIR + Medfilt + SVR, and (**g**) total anthocyanins by TEX + S-G + SVR.

**Table 1 foods-14-02938-t001:** The application of nondestructive testing technology in tea grading and quality evaluation.

Tea	Research Target	Technology	Machine Learning Algorithm	Reference
Vine tea	Chemical composition (myricetin, dihydromyricetin, quercetin, kaempferol, and quercitrin.)	LC-MS, GC-MS	PCA, RF	[12]
Dark tea	Classification of black tea at different altitudes	HPLC, DAD, ELSD	HPLC, HPLC-DAD, HPLC-ELSD	[10]
Keemun black tea	Classification of black tea, quantitative prediction of the concentrations of chemical components (GA, CAFF, EGC, C, EGCG, EC, GCG, total catechins)	micro-NIR, CV, CAS	SVM, LS-SVM, ELM, PLS-DA	[13]
Oolong tea	Variety classification	Gustative sensor system, CSA	PCA, LDA, CA, ANN	[14]
Black tea	Grade evaluation	Hyperspectral	PLS-DA, SVM, PNN	[15]
White tea	Grade evaluation, chemical composition (catechins, tea polyphenols, and free amino acids), sensory evaluation	Color texture feature, hyperspectral	SVM, KNN	[9]

**Table 2 foods-14-02938-t002:** Abbreviations of different data fusion methods and corresponding instructions.

Abbreviation	Instructions
NIR	Spectral data from 900 to 1700 nm
TEX	Nine color feature factors and five texture feature factors
NIR + TEX	The fusion data of NIR and TEX

**Table 3 foods-14-02938-t003:** Distribution of total anthocyanin content across tenderness levels.

Level	Range (mg/g)	Mean	STD	CV
Level 1	17.498–18.303	17.931	0.211	0.012
Level 2	18.215–18.998	18.668	0.241	0.013
Level 3	19.477–20.120	19.783	0.180	0.009
Level 4	19.665–20.325	19.963	0.217	0.011
Level 5	14.965–15.843	15.377	0.262	0.017

**Table 4 foods-14-02938-t004:** Performance metrics of the tenderness level prediction model for Zijuan tea fresh leaves implementing PSO-SVM algorithm.

Model	Data Fusion Methods	Preprocessing Method	PCs	Calibration Set		Prediction Set	
				Result	CCR	Result	CCR
PSO-SVM	NIR	Raw	12	74/80	92.50%	16/20	80.00%
		S-G	11	74/80	92.50%	16/20	80.00%
		SNV	10	65/80	81.50%	15/20	75.00%
		Medfilt	16	77/80	96.50%	18/20	90.00%
		Normaliz	11	71/80	88.75%	14/20	70.00%
	TEX	Raw	10	80/80	100.00%	9/20	45.00%
		S-G	10	80/80	100.00%	15/20	75.00%
		SNV	8	58/80	72.50%	8/20	40.00%
		Medfilt	10	80/80	100.00%	18/20	90.00%
		Normaliz	18	62/80	77.50%	8/20	40.00%
	NIR + TEX	Raw	18	20/80	25.00%	0/20	0.00%
		S-G	19	78/80	97.50%	13/20	65.00%
		SNV	21	66/80	82.50%	12/20	60.00%
		Medfilt	9	80/80	100.00%	17/20	85.00%
		Normaliz	19	62/80	77.50%	10/20	50.00%

**Table 5 foods-14-02938-t005:** Performance metrics of the tenderness level prediction model for Zijuan tea fresh leaves implementing RF algorithm.

Model	Data Fusion Methods	Preprocessing Method	PCs	Calibration Set		Prediction Set	
				Result	CCR	Result	CCR
RF	NIR	Raw	12	80/80	100.00%	15/20	75.00%
		S-G	10	80/80	100.00%	18/20	90.00%
		SNV	11	80/80	100.00%	19/20	95.00%
		Medfilt	11	80/80	100.00%	19/20	95.00%
		Normaliz	11	80/80	100.00%	16/20	80.00%
	TEX	Raw	10	80/80	100.00%	17/20	85.00%
		S-G	12	80/80	100.00%	15/20	75.00%
		SNV	8	80/80	100.00%	12/20	60.00%
		**Medfilt**	**10**	**80/80**	**100.00%**	**20/20**	**100.00%**
		Normaliz	8	80/80	100.00%	11/20	55.00%
	NIR + TEX	Raw	19	80/80	100.00%	15/20	75.00%
		S-G	15	80/80	100.00%	14/20	70.00%
		SNV	21	80/80	100.00%	16/20	80.00%
		Medfilt	9	80/80	100.00%	17/20	85.00%
		Normaliz	19	80/80	100.00%	13/20	65.00%

**Table 6 foods-14-02938-t006:** Performance metrics of the tenderness level prediction model for Zijuan tea fresh leaves implementing CNN algorithm.

Model	Data Fusion Methods	Preprocessing Method	PCs	Calibration Set		Prediction Set	
				Result	CCR	Result	CCR
CNN	NIR	Raw	12	80/80	100.00%	17/20	85.00%
		S-G	10	80/80	100.00%	19/20	95.00%
		SNV	14	80/80	100.00%	19/20	95.00%
		**Medfilt**	**11**	**80/80**	**100.00%**	**20/20**	**100.00%**
		Normaliz	11	80/80	100.00%	13/20	65.00%
	TEX	Raw	10	79/80	98.75%	12/20	60.00%
		S-G	12	80/80	100.00%	9/20	45.00%
		SNV	8	80/80	100.00%	9/20	45.00%
		Medfilt	10	78/80	97.50%	17/20	85.00%
		Normaliz	8	80/80	100.00%	10/20	50.00%
	NIR + TEX	Raw	21	80/80	100.00%	15/20	75.00%
		S-G	15	80/80	100.00%	15/20	75.00%
		SNV	21	80/80	100.00%	15/20	75.00%
		Medfilt	25	80/80	100.00%	19/20	95.00%
		Normaliz	19	80/80	100.00%	13/20	65.00%

**Table 7 foods-14-02938-t007:** Model performance metrics for delphinidin-3-O-galactoside under triple-data-fusion strategies and fourfold preprocessing methods.

Component	Data Fusion Method	Preprocessing Method	PCs	Calibration Set		Prediction Set		RPD
				Rc	RMSEC (mg/g)	Rp	RMSEP (mg/g)	
Delphinidin-3-O-galactoside	NIR	Raw	16	0.99	0.01	0.77	0.10	1.441
		S-G	17	0.97	0.07	0.80	0.10	1.528
		SNV	15	0.97	0.07	0.79	0.12	1.540
		Medfilt	10	0.99	0.03	0.95	0.08	2.776
		Normaliz	16	0.91	0.11	0.85	0.12	1.804
	TEX	Raw	5	0.94	0.09	0.90	0.12	2.214
		S-G	5	0.97	0.06	0.97	0.08	4.072
		SNV	8	0.99	0.04	0.90	0.10	2.244
		**Medfilt**	**5**	**0.98**	**0.06**	**0.98**	**0.06**	**4.687**
		Normaliz	6	0.97	0.07	0.92	0.09	2.588
	NIR + TEX	Raw	24	0.98	0.06	0.91	0.13	2.032
		S-G	16	0.98	0.06	0.90	0.12	2.287
		SNV	21	0.97	0.07	0.89	0.12	2.145
		Medfilt	7	0.97	0.06	0.96	0.08	3.368
		Normaliz	26	0.99	0.01	0.89	0.12	1.957

**Table 8 foods-14-02938-t008:** Model performance metrics for delphinidin-3-O-glucoside under triple-data-fusion strategies and fourfold preprocessing methods.

Component	Data Fusion Method	Preprocessing Method	PCs	Calibration Set		Prediction Set		RPD
				Rc	RMSEC (mg/g)	Rp	RMSEP (mg/g)	
Delphinidin-3-O-glucoside	NIR	Raw	16	0.99	0.02	0.51	0.10	1.111
		S-G	13	0.99	0.02	0.51	0.10	1.111
		SNV	13	0.96	0.04	0.89	0.05	2.158
		Medfilt	9	0.99	0.01	0.88	0.06	2.082
		Normaliz	13	0.98	0.03	0.71	0.09	1.421
	TEX	Raw	5	0.87	0.07	0.82	0.10	1.525
		**S-G**	**9**	**0.94**	**0.05**	**0.96**	**0.05**	**3.718**
		SNV	8	0.96	0.04	0.88	0.06	2.109
		Medfilt	7	0.97	0.03	0.95	0.05	2.992
		Normaliz	6	0.93	0.06	0.86	0.07	1.973
	NIR + TEX	Raw	18	0.99	0.02	0.82	0.10	1.545
		S-G	15	0.98	0.03	0.87	0.08	1.948
		SNV	17	0.94	0.05	0.78	0.09	1.570
		Medfilt	8	0.98	0.03	0.93	0.06	2.616
		Normaliz	17	0.98	0.03	0.82	0.09	1.636

**Table 9 foods-14-02938-t009:** Model performance metrics for cyanidin-3,5-O-diglucoside under triple-data-fusion strategies and fourfold preprocessing methods.

Component	Data Fusion Method	Preprocessing Method	PCs	Calibration Set		Prediction Set		RPD
				Rc	RMSEC (mg/g)	Rp	RMSEP (mg/g)	
Cyanidin-3,5-O-diglucoside	NIR	Raw	9	0.97	0.05	0.75	0.08	1.328
		S-G	9	0.97	0.05	0.77	0.07	1.471
		SNV	7	0.94	0.07	0.80	0.08	1.680
		Medfilt	6	0.97	0.05	0.91	0.07	2.410
		Normaliz	7	0.96	0.05	0.85	0.08	1.823
	TEX	Raw	7	0.97	0.04	0.91	0.09	2.294
		S-G	10	0.96	0.05	0.94	0.07	2.755
		SNV	7	0.90	0.08	0.84	0.10	1.628
		Medfilt	7	0.89	0.08	0.87	0.09	2.026
		Normaliz	7	0.92	0.08	0.84	0.10	1.657
	NIR + TEX	Raw	17	0.95	0.06	0.92	0.08	2.510
		S-G	14	0.99	0.03	0.92	0.09	2.439
		SNV	15	0.93	0.07	0.91	0.07	2.442
		**Medfilt**	**7**	**0.92**	**0.07**	**0.93**	**0.07**	**2.736**
		Normaliz	16	0.97	0.05	0.90	0.07	2.242

**Table 10 foods-14-02938-t010:** Model performance metrics for cyanidin-3-O-glucoside under triple-data-fusion strategies and fourfold preprocessing methods.

Component	Data Fusion Method	Preprocessing Method	PCs	Calibration Set		Prediction Set		RPD
				Rc	RMSEC (mg/g)	Rp	RMSEP (mg/g)	
Cyanidin-3-O-glucoside	NIR	Raw	6	0.94	0.11	0.91	0.10	2.396
		S-G	11	0.95	0.10	0.88	0.11	2.157
		SNV	6	0.92	0.13	0.91	0.10	2.214
		**Medfilt**	**4**	**0.95**	**0.10**	**0.95**	**0.08**	**3.267**
		Normaliz	11	0.95	0.11	0.94	0.09	2.969
	TEX	Raw	10	0.91	0.12	0.85	0.16	1.877
		S-G	10	0.97	0.07	0.94	0.09	2.866
		SNV	8	0.95	0.09	0.85	0.15	1.739
		Medfilt	10	0.94	0.11	0.90	0.12	2.319
		Normaliz	7	0.97	0.07	0.84	0.16	1.795
	NIR + TEX	Raw	16	0.96	0.08	0.93	0.12	2.405
		S-G	14	0.99	0.02	0.94	0.11	2.865
		SNV	23	0.98	0.07	0.89	0.13	2.120
		Medfilt	9	0.98	0.06	0.94	0.11	2.815
		Normaliz	24	0.99	0.02	0.90	0.12	2.227

**Table 11 foods-14-02938-t011:** Model performance metrics for petunidin under triple-data-fusion strategies and fourfold preprocessing methods.

Component	Data Fusion Method	Preprocessing Method	PCs	Calibration Set		Prediction Set		RPD
				Rc	RMSEC (mg/g)	Rp	RMSEP (mg/g)	
Petunidin	NIR	Raw	9	0.97	0.25	0.76	0.54	1.525
		S-G	10	0.99	0.17	0.81	0.50	1.648
		SNV	8	0.98	0.18	0.87	0.45	2.040
		**Medfilt**	**8**	**0.97**	**0.23**	**0.91**	**0.42**	**2.465**
		Normaliz	9	0.98	0.21	0.86	0.48	2.000
	TEX	Raw	6	0.83	0.56	0.51	0.89	1.082
		S-G	10	0.98	0.20	0.89	0.53	2.023
		SNV	7	0.81	0.62	0.56	0.81	1.090
		Medfilt	10	0.99	0.04	0.84	0.62	1.706
		Normaliz	8	0.91	0.43	0.55	0.83	1.190
	NIR + TEX	Raw	16	0.95	0.31	0.74	0.64	1.385
		S-G	21	0.99	0.16	0.79	0.65	1.528
		SNV	15	0.96	0.31	0.77	0.61	1.545
		Medfilt	12	0.92	0.39	0.75	0.69	1.495
		Normaliz	14	0.92	0.41	0.75	0.65	1.387

**Table 12 foods-14-02938-t012:** Model performance metrics for cyanidin under triple-data-fusion strategies and fourfold preprocessing methods.

Component	Data Fusion Method	Preprocessing Method	PCs	Calibration Set		Prediction Set		RPD
				Rc	RMSEC (mg/g)	Rp	RMSEP (mg/g)	
Cyanidin	NIR	Raw	7	0.75	0.11	0.61	0.08	1.246
		S-G	12	0.96	0.05	0.68	0.07	1.320
		SNV	8	0.81	0.10	0.70	0.09	1.359
		**Medfilt**	**4**	**0.93**	**0.05**	**0.90**	**0.07**	**2.304**
		Normaliz	13	0.78	0.10	0.41	0.12	1.103
	TEX	Raw	5	0.69	0.11	0.40	0.15	1.055
		S-G	13	0.94	0.05	0.50	0.15	1.149
		SNV	6	0.74	0.11	0.68	0.10	1.333
		Medfilt	10	0.70	0.11	0.64	0.12	1.168
		Normaliz	7	0.83	0.09	0.88	0.07	2.012
	NIR + TEX	Raw	7	0.80	0.09	0.41	0.14	0.987
		S-G	8	0.99	0.02	0.73	0.10	1.488
		SNV	8	0.81	0.09	0.62	0.11	1.285
		Medfilt	7	0.85	0.08	0.66	0.11	1.272
		Normaliz	8	0.99	0.01	0.87	0.08	1.857

**Table 13 foods-14-02938-t013:** Model performance metrics for total anthocyanins under triple-data-fusion strategies and fourfold preprocessing methods.

Component	Data Fusion Method	Preprocessing Method	PCs	Calibration Set		Prediction Set		RPD
				Rc	RMSEC (mg/g)	Rp	RMSEP (mg/g)	
Total anthocyanins	NIR	Raw	10	0.97	0.34	0.65	0.74	1.245
		S-G	10	0.99	0.22	0.72	0.64	1.434
		SNV	8	0.94	0.50	0.87	0.53	1.980
		Medfilt	6	0.98	0.26	0.95	0.43	2.979
		Normaliz	10	0.99	0.22	0.85	0.60	1.860
	TEX	Raw	5	0.91	0.55	0.82	0.84	1.694
		**S-G**	**10**	**0.97**	**0.28**	**0.96**	**0.49**	**3.280**
		SNV	6	0.88	0.68	0.76	0.81	1.505
		Medfilt	5	0.96	0.37	0.94	0.51	2.749
		Normaliz	6	0.91	0.60	0.82	0.69	1.753
	NIR + TEX	Raw	23	0.99	0.08	0.91	0.76	1.818
		S-G	20	0.99	0.04	0.85	0.89	1.639
		SNV	24	0.99	0.12	0.89	0.66	1.945
		Medfilt	11	0.94	0.46	0.86	0.75	1.949
		Normaliz	24	0.99	0.22	0.87	0.67	1.769

**Table 14 foods-14-02938-t014:** Model performance metrics for petunidin waveband selection.

Component	Optimal Combination	Method	Variable Number	PCs	Calibration Set		Prediction Set		RPD
					Rc	RMSEC (mg/g)	Rp	RMSEP (mg/g)	
Petunidin	NIR + Medfilt	Raw	751	8	0.97	0.23	0.91	0.42	2.465
		CARS	52	11	0.99	0.04	0.93	0.40	2.584
		BOSS	20	8	0.99	0.03	0.89	0.49	1.995
		**SPA**	**37**	**5**	**0.97**	**0.25**	**0.95**	**0.34**	**2.888**

**Table 15 foods-14-02938-t015:** Performance metrics for linear versus nonlinear predictive models of total anthocyanin content and key components.

Component	Optimal Combination	Model	PCs	Calibration Set		Prediction Set		RPD
				Rc	RMSEC (mg/g)	Rp	RMSEP (mg/g)	
Delphinidin-3-O-galactoside	TEX + Medfilt	PLSR	5	0.89	0.12	0.95	0.08	3.135
		**SVR**	**5**	**0.98**	**0.06**	**0.98**	**0.06**	**4.687**
		RF	5	0.99	0.06	0.97	0.09	3.726
Delphinidin-3-O-glucoside	TEX + S-G	PLSR	9	0.92	0.05	0.96	0.05	3.567
		**SVR**	**9**	**0.94**	**0.05**	**0.96**	**0.05**	**3.718**
		RF	9	0.98	0.04	0.96	0.07	2.901
Cyanidin-3,5-O-diglucoside	TEX + S-G	PLSR	10	0.83	0.09	0.94	0.07	2.850
		**SVR**	**10**	**0.96**	**0.05**	**0.94**	**0.07**	**2.755**
		RF	10	0.98	0.05	0.94	0.11	2.438
Cyanidin-3-O-glucoside	NIR + Medfilt	PLSR	4	0.93	0.11	0.93	0.10	2.629
		**SVR**	**4**	**0.95**	**0.10**	**0.95**	**0.08**	**3.267**
		RF	4	0.98	0.07	0.93	0.11	3.532
Petunidin	NIR + Medfilt + SPA	PLSR	5	0.47	0.90	0.72	0.71	0.958
		**SVR**	**5**	**0.97**	**0.25**	**0.95**	**0.34**	**2.888**
		RF	5	0.98	0.30	0.92	0.44	2.573
Cyanidin	NIR + Medfilt	PLSR	4	0.15	0.15	0.55	0.13	0.334
		**SVR**	**4**	**0.93**	**0.05**	**0.90**	**0.07**	**2.304**
		RF	4	0.96	0.05	0.65	0.11	1.893
Total anthocyanins	TEX + S-G	PLSR	10	0.89	0.57	0.94	0.55	2.732
		**SVR**	**10**	**0.97**	**0.28**	**0.96**	**0.49**	**3.280**
		RF	10	0.99	0.37	0.95	0.85	2.455

## Data Availability

The original contributions presented in the study are included in the article, further inquiries can be directed to the corresponding authors.

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
