# Peer review of "Assessment of Tenderness and Anthocyanin Content in Zijuan Tea Fresh Leaves Using Near-Infrared Spectroscopy Fused with Visual Features"

_foods, 2025, doi:10.3390/foods14172938_

Round 1
Reviewer 1 Report
Comments and Suggestions for Authors
-
Explain the new idea of your study more clearly.
-
Add more recent and related references.
-
Show more details of your methods.
-
Explain how you fused NIR and image data step by step.
-
Use a bigger sample size or use a simpler model.
-
Add validation with new or outside data.
-
Show error bars and p-values in your figures.
-
Improve the captions of your tables and figures.
-
Make the results easier to understand.
-
Use simpler and clearer English in some parts.
-
Most parts are understandable.
-
Some sentences are too long or unclear.
-
Grammar mistakes appear in several places.
-
Word choices can be improved.
-
Please ask a native speaker or language editor to check.
Author Response
Comments 1: Explain the new idea of your study more clearly.
Response 1: Thank you very much for your comments. We fully agree with this suggestion and have revised the Introduction to better clarify the innovative aspects of this study.“ This study is based on the unique characteristics of the Zijuan tea variety and innovatively integrates near-infrared spectroscopy with visual features such as color and texture of fresh Zijuan tea leaves, aiming to establish models for tenderness classification and key anthocyanin content detection.” Meanwhile, we drew the graphical abstract of this study. Express the research ideas of this study simply and clearly.(Lns. 92-95; Figure 1)
Comments 2: Add more recent and related references.
Response 2: Thank you for your valuable suggestions. We have incorporated several recently published relevant references into the manuscript. Additionally, a comparative table summarizing the current research status of Zijuan tea has been inserted in the Introduction section to provide a systematic overview.(Table 1)
Comments 3: Show more details of your methods.
Response 3: Thank you for your valuable suggestions. We have significantly expanded the content of each subsection in the "Materials and Methods" section to provide greater methodological detail. Additionally, a graphical abstract illustrating the core experimental workflow has been included as Figure 1 in the revised manuscript.(Figure 1)
Comments 4: Explain how you fused NIR and image data step by step.
Response 4: Thank you for your constructive feedback. In Section 2.5 "Data Fusion Strategies," we provide a comprehensive explanation of data fusion methodologies, which are conventionally categorized into three hierarchical levels: low-level, mid-level, and high-level. Specifically, low-level data fusion involves the direct concatenation of raw data from heterogeneous sources for subsequent modeling. However, this approach retains excessive noise and redundant information, which may impair model performance. In contrast, mid-level data fusion performs preprocessing and feature selection (e.g., wavelength selection) on concatenated features. This method extracts discriminative characteristics from multi-modal data while preserving critical information and eliminating significant redundancy. High-level data fusion, also termed decision-level fusion, involves complex strategy design for integrating processed outcomes. Therefore, this study employs mid-level data fusion to balance information integrity and computational efficiency. (Lns. 210-215)
Comments 5: Use a bigger sample size or use a simpler model.
Response 5: Thank you for your valuable suggestions. We have comprehensively considered machine learning modeling approaches, where sample size relative to feature dimensionality typically influences model performance. Optimal results are generally achieved when the sample size exceeds ten times the number of features. In this study, both the optimized Zijuan tea tenderness classification model and the key anthocyanin content prediction model utilized fewer than 10 principal components after PCA dimensionality reduction. With a sample size of 100, our design satisfies this critical requirement. The implemented linear PLSR and nonlinear SVR and RF models represent robust machine learning methodologies with high predictive and classification performance. All optimized models demonstrated computational efficiency by completing operations within one second during testing, thereby establishing a solid computational foundation for developing intelligent assembly-line equipment for Zijuan tea fresh-leaf processing.
Comments 6: Add validation with new or outside data.
Response 6: Thank you for your constructive suggestions. This study established a tenderness classification model and a key anthocyanin content prediction model for Zijuan tea fresh leaves, utilizing a total of 100 samples. Prior to modeling, the dataset was partitioned in an 80:20 ratio using the Kennard-Stone (K-S) algorithm, with 80 samples allocated for model training and the remaining 20 reserved as an independent test set. Evaluation metrics confirm that all anthocyanin prediction models delivered satisfactory performance: high and closely aligned RMSEC (training set) and RMSEV (test set) values indicate minimal overfitting or underfitting, while low RMSEC and RMSEP values further validate model robustness. These optimized models establish a solid computational foundation for developing intelligent equipment to assess Zijuan tea fresh-leaf tenderness and anthocyanin levels.
Comments 7: Show error bars and p-values in your figures.
Response 7: Thank you very much for your revision suggestions. Both Figures 2 and 3 of this study are violin plots. The essence of a violin plot is to visualize the data distribution. Its advantage lies in the combination of kernel density estimation (KDE) and box plots. The median and quartiles (25% and 75%) of the key anthocyanin contents are displayed through the box plot section in the middle, and the density distribution of the data is shown through the symmetrical "violin" shape on both sides. The violin chart pays more attention to the distribution of data rather than the "mean ± standard deviation". In Table 2, data statistics were conducted for the anthocyanin contents of different grades, presenting the average value, standard deviation and coefficient of variation of the data. In this study, a correlation analysis was conducted among the five different grades of each key anthocyanin. Different symbols were used in Figures 2 and 3 to represent different P-values, where ns represents p>0.05, * represents 0.01<p≤0.05, ** represents 0.001<p≤0.01, and *** represents p<0.001. If the specific P-values between every two contents were displayed in the graph, it would make the entire graph overly complicated. Therefore, a more concise and intuitive form was adopted to express the P-values.
Comments 8: Improve the captions of your tables and figures.
Response 8: Thank you very much for your revision suggestions. We have made detailed improvements to the titles of each figure and table, making the language more concise and the expression clearer. (Ln.90; Ln.119; Ln.216; Ln.298; Ln.300; Ln.316; Ln.339; Ln.384; Ln.388; Ln.391; Ln.393; Ln.395; Ln.431; Ln.433; Ln.435; Ln.437; Ln.439; Ln.441; Ln.443; Ln.464; Ln.466; Ln.486; Ln.489)
Comments 9: Make the results easier to understand.
Response 9: Thank you very much for your valuable suggestions. We revised the third section "Results and Analysis", and improved the result analysis of each subsection to make the results easier to understand.
Comments 10: Use simpler and clearer English in some parts.
Response 10: Thank you very much for your valuable comments on the language of this research. We have reorganized the English text of the entire text.
Reviewer 2 Report
Comments and Suggestions for Authors
This manuscript presents a well-designed and scientifically sound study that integrates near-infrared spectroscopy (NIR) with visual features to classify leaf tenderness and predict anthocyanin content in Zijuan tea. The research is timely, methodologically rigorous, and offers practical value for intelligent tea quality assessment.
The introduction provides sufficient background and includes relevant references to support the study's context and rationale, but it could be enhanced by: briefly summarizing previous modeling approaches (e.g., PLSR, SVR) used in similar contexts.
While the design is strong, the paper could benefit from:
- A clearer explanation of why specific models (e.g., CNN vs. RF) were chosen for certain tasks.
- More detail on how visual features were standardized across samples (e.g., lighting conditions).
The methods section is comprehensive, reproducible, and well-structured, making it easy for other researchers to replicate the study or build upon it.
Some tables (e.g., Tables 6–14) are dense; a summary table highlighting the best-performing models could improve readability or marked by red the best-performing models in the current tables.
A brief graphical abstract or flowchart summarizing the modeling pipeline might help readers grasp the workflow more quickly.
The conclusions are well-supported by the results presented in the manuscript. Moreover the conclusions are not only consistent with the data but also appropriately cautious where needed (e.g., noting overfitting in some models). The authors avoid overgeneralization and base their claims on quantitative evidence.

Author Response
Comments 1: The introduction provides sufficient background and includes relevant references to support the study's context and rationale, but it could be enhanced by: briefly summarizing previous modeling approaches (e.g., PLSR, SVR) used in similar contexts.
Response 1: Thank you very much for your suggestions for revision on this research. We drew Table 1, briefly summarizing the application of machine learning in the field of tea. (Table 1)
Comments 2: A clearer explanation of why specific models (e.g., CNN vs. RF) were chosen for certain tasks.
Response 2: Thank you very much for your revision suggestions. In Sections 2.6 and 2.7, we briefly outlined the advantages of different models selected for different tasks.(Lns.222-230; Lns.252-258)
Comments 3: More detail on how visual features were standardized across samples (e.g., lighting conditions).
Response 3: Thank you very much for your suggestions for revision. We detailed the methods of visual features and data fusion in Section 2.5. We provide a comprehensive explanation of data fusion methodologies, which are conventionally categorized into three hierarchical levels: low-level, mid-level, and high-level. Specifically, low-level data fusion involves the direct concatenation of raw data from heterogeneous sources for subsequent modeling. However, this approach retains excessive noise and redundant information, which may impair model performance. In contrast, mid-level data fusion performs preprocessing and feature selection (e.g., wavelength selection) on concatenated features. This method extracts discriminative characteristics from multi-modal data while preserving critical information and eliminating significant redundancy. High-level data fusion, also termed decision-level fusion, involves complex strategy design for integrating processed outcomes. Therefore, this study employs mid-level data fusion to balance information integrity and computational efficiency.
Comments 4: The methods section is comprehensive, reproducible, and well-structured, making it easy for other researchers to replicate the study or build upon it.
Response 4: Thank you very much for your affirmation of the methodology of this research content.
Comments 5: Some tables (e.g., Tables 6–14) are dense; a summary table highlighting the best-performing models could improve readability or marked by red the best-performing models in the current tables.
Response 5: Thank you very much for your constructive suggestions on the table content of this study. We have improved all the tables and represented all the optimal models in bold font.
Comments 6: A brief graphical abstract or flowchart summarizing the modeling pipeline might help readers grasp the workflow more quickly.
Response 6: Thank you very much for your revision suggestions. We have drawn a graphic summary. Please refer to Figure 1. (Figure 1)
Comments 7: The conclusions are well-supported by the results presented in the manuscript. Moreover the conclusions are not only consistent with the data but also appropriately cautious where needed (e.g., noting overfitting in some models). The authors avoid overgeneralization and base their claims on quantitative evidence.
Response 7: Thank you very much for your affirmation of the conclusion part of this research.
Reviewer 3 Report
Comments and Suggestions for Authors
Dear Author please find some suggestions,
The study targets Zijuan tea, known for its rich anthocyanin content, which is relatively underexplored in the context of quality evaluation and grading systems.
The fusion of near-infrared (NIR) spectroscopy with visual features for evaluating tea leaf tenderness and anthocyanin content is a novel methodological combination, enhancing classification accuracy and robustness.
The use of optimized models like PSO-SVM, CNN, and RF for tenderness classification, along with both linear (PLSR) and nonlinear (SVR, RF) models for content prediction, reflects a novel and comprehensive modeling strategy.
Achieving 100% classification accuracy and highly accurate anthocyanin content prediction is supporting the approach and solution for real-time quality control in tea production.
A table of comparison is worth to include in the Ms to present the importance and advancement of the methodology over other approaches.
The Research should also comment on the exploring the generalizability of the proposed method across different tea cultivars and harvesting conditions to ensure broader applicability.
Furthermore, the dataset and deep learning model needs, more diverse samples to enhance robustness.
Validating the models in an industrial setting with continuous feedback from tea experts would strengthen practical relevance.
Author Response
Comments 1: The study targets Zijuan tea, known for its rich anthocyanin content, which is relatively underexplored in the context of quality evaluation and grading systems.
Response 1: We sincerely appreciate your suggestions. Camellia sinensis cv. 'Zijuan'—a cultivated tea variety with genetically elevated anthocyanin levels—exhibits significantly higher anthocyanin content compared to conventional tea cultivars. Given the prohibitively high costs associated with current gold-standard detection methods (e.g., High-Performance Liquid Chromatography [HPLC] and Liquid Chromatography-Mass Spectrometry [LC-MS]) for precise anthocyanin quantification, this study pioneers the application of spectral and computer vision-based non-destructive detection techniques to enable rapid, non-destructive quantification of anthocyanins in Zijuan tea leaves.
Comments 2: The fusion of near-infrared (NIR) spectroscopy with visual features for evaluating tea leaf tenderness and anthocyanin content is a novel methodological combination, enhancing classification accuracy and robustness.
Response 2: Thank you very much for your recognition of this research.
Comments 3: The use of optimized models like PSO-SVM, CNN, and RF for tenderness classification, along with both linear (PLSR) and nonlinear (SVR, RF) models for content prediction, reflects a novel and comprehensive modeling strategy.
Response 3: Thank you very much for your recognition of this research.
Comments 4: Achieving 100% classification accuracy and highly accurate anthocyanin content prediction is supporting the approach and solution for real-time quality control in tea production.
Response 4: Thank you very much for your recognition of this research.
Comments 5: A table of comparison is worth to include in the Ms to present the importance and advancement of the methodology over other approaches.
Response 5: Thank you very much for your revision suggestions. We added Table 1 in the "Introduction" of the manuscript, briefly summarizing the application of non-destructive testing technology in the field of tea. Among them, some researchers have adopted a single technology, while others have integrated multiple technologies, all of which can achieve the evaluation of tea quality. (Table 1)
Comments 6: The Research should also comment on the exploring the generalizability of the proposed method across different tea cultivars and harvesting conditions to ensure broader applicability.
Response 6: We greatly appreciate your revision suggestions. This study exclusively focuses on quality evaluation and tenderness grade discrimination for tea. 'Zijuan' fresh leaves, providing a theoretical foundation for developing intelligent grading equipment targeting raw material at the initial processing stage. Following rapid non-destructive identification of anthocyanin content in Zijuan tea leaves via our designed device, subsequent processing (e.g., green tea, black tea, white tea) can be customized based on anthocyanin levels and enterprise requirements.
Comments 7: Furthermore, the dataset and deep learning model needs, more diverse samples to enhance robustness.
Response 7: Thank you very much for your revision suggestions. We comprehensively considered machine learning modeling methods. The sample size and the number of features usually affect the performance of the model. Generally speaking, the model performs best when the sample size is more than ten times the number of features. In the optimal tenderness discrimination model and key anthocyanin content prediction model of Zijuan tea established in this study, the number of features after PCA dimensionality reduction is both below 10. The sample size used in this study is 100, which conforms to this characteristic. Linear PLSR, nonlinear SVR and RF models are all powerful machine learning modeling methods with high prediction and classification performance. After testing, the optimal prediction models established in this study can all be completed within one second, providing a solid theoretical foundation for the subsequent development of intelligent equipment for the fresh leaf production line of Zijuan tea.
Comments 8: Validating the models in an industrial setting with continuous feedback from tea experts would strengthen practical relevance.
Response 8: Thank you very much for your revision suggestions. After the development of our intelligent grading equipment and quality evaluation equipment is completed, during the actual application period, the error between the actual anthocyanin content and the anthocyanin content output by this equipment will be regularly measured, thereby iteratively optimizing the performance of this equipment.
Reviewer 4 Report
Comments and Suggestions for Authors
I have reviewed the manuscript entitled “Assessment of Tenderness and Anthocyanin Content in Zijuan Tea Fresh Leaves Using Near-Infrared Spectroscopy Fused with 3 Visual Features” which proposes a rapid and non-destructive method for evaluating tenderness and anthocyanin content in fresh Zijuan tea leaves. The paper is clearly structured and well-written. The following comments are in order to improve the manuscript:
Introduction Section. The introduction is well-written and provides the necessary background for understanding the scope and relevance of the study.
Materials and Methods Section.
-Section 2.1: Please consider including an image of the tea leaf samples to visually represent the different levels of the fresh Zijuan leaves.
-Please standardize the number of significant figures in the reported volumes in lines 121, 126, 127, and 129.
-Check the spacing between the numbers and the unit °C in lines 142 and 143.
-Review the bibliographic formatting and citation style in lines 175 and 176.
Results and Analysis Section.
-Although not the main objective of the study, it would be helpful to expand on the relationship between anthocyanin content and leaf maturity level.
-How does the anthocyanin content reported in this study compare with values from other studies in the literature?
-Figure 1b: The image is not very clear in showing the differences between levels 1–5. Please consider whether a table might better present this information.
-Figure 7: Please enlarge the figure to improve readability and interpretation.
Author Response
Comments 1: Introduction Section. The introduction is well-written and provides the necessary background for understanding the scope and relevance of the study.
Response 1: Thank you very much for your recognition of this research.
Comments 2: Section 2.1: Please consider including an image of the tea leaf samples to visually represent the different levels of the fresh Zijuan leaves.
Response 2: Thank you very much for your suggestions for revision. In Section 2.1, we added a graphical abstract of this study, which includes images of purple rhododendron tea samples and a simplified diagram of the overall research approach of this study. (Figure 1)
Comments 3: Please standardize the number of significant figures in the reported volumes in lines 121, 126, 127, and 129.
Response 3: Thank you very much for your revision suggestions. Our opinions have standardized the numbers in this subsection, and now they are all displayed in integer form. (Ln.132; Ln.138; Ln.140)
Comments 4: Check the spacing between the numbers and the unit °C in lines 142 and 143.
Response 4: Thank you very much for your valuable suggestions for revision. We have added Spaces between the numbers and the degrees Celsius. (Ln.125; Ln.136; Ln.150; Ln.153; Ln.154; Ln.165; Ln.177)
Comments 5: Review the bibliographic formatting and citation style in lines 175 and 176.
Response 5: Thank you very much for your suggestions for improvement. The references here have been corrected. Format: "last name +et al., year". (Ln.189; Ln.194; Ln.197)
Comments 6: Although not the main objective of the study, it would be helpful to expand on the relationship between anthocyanin content and leaf maturity level.
Response 6: Section 3.1 provides a detailed analysis of the differential distribution of 16 anthocyanin compounds across various tenderness grades of Zijuan fresh leaves. Our investigation revealed a stepwise increase in total anthocyanin content from Grade 1 to Grade 4, followed by a statistically significant reduction to minimal levels in Grade 5 (fully expanded mature leaves).(Lns.281-282)
Comments 7: How does the anthocyanin content reported in this study compare with values from other studies in the literature?
Response 7: Thank you very much for your revision suggestions. We referred to the relevant literature on Zijuan tea. The Zijuan tea samples used in this study were sourced from the fresh spring tea leaves at the base of the Tea Research Institute of the Chinese Academy of Agricultural Sciences in Hangzhou, Zhejiang Province, China. Chen et al. analyzed the anthocyanin content of purple rhododendron tea in different regions of Yunnan and found that the total anthocyanin content ranged from 357.91 to 690.19μg/g. (Chen et al., 2023) Jiang et al. analyzed the anthocyanin content by HPLC to be 707μg/g, and 76.55mg/g in spring. (Jiang et al., 2013) The amount of anthocyanins in the tea made from Sunrouge was 0.86-2.17 mg/g (Nesumi et al, 2012). The anthocyanin content of the Zijuan tea in this study was 14.965-20.325mg/g. There are significant differences in the accumulation of anthocyanin content among different varieties of Zijuan tea. This difference might be caused by various factors such as tea tree varieties, growth conditions, detection methods, and sample preparation (sampling time and location, etc.). The fresh leaves of the spring Zijuan tea in Hangzhou, China were picked in this study. Due to the complexity and variability of environmental conditions, the regulatory mechanism of anthocyanin content in the fresh Zijuan tea leaves in different regions still needs further research.
Comments 8: Figure 1b: The image is not very clear in showing the differences between levels 1–5. Please consider whether a table might better present this information.
Response 8: Thank you very much for your revision suggestions. The trace anthocyanin content values in Figure 1b are relatively low, and the value differences expressed through the violin plot are not significant. Therefore, we have added a table to clearly display the range, mean, standard deviation, and coefficient of variation of trace anthocyanin content for different tenderness grades. Considering that too many values make the Table too large, we add it as Table S1. (Table S1)
Comments 9: Figure 7: Please enlarge the figure to improve readability and interpretation.
Response 9: Thank you very much for your suggestions for revision. We have enlarged the font size of this figure.
Reference
Chen, Y., J. Yang, Q. Meng, and H. Tong. "Non-Volatile Metabolites Profiling Analysis Reveals the Tea Flavor of "Zijuan" in Different Tea Plantations." Food Chem 412 (Jun 30 2023): 135534.
Jiang, L., X. Shen, T. Shoji, T. Kanda, J. Zhou, and L. Zhao. "Characterization and Activity of Anthocyanins in Zijuan Tea (Camellia Sinensis Var. Kitamura)." J Agric Food Chem 61, no. 13 (Apr 3 2013): 3306-10.
Nesumi, A., Ogino, A., Yoshida, K., Taniguchi, F., Maeda Yamamoto, M., Tanaka, J., & Murakami, A. (2012). ‘Sunrouge’, a new tea cultivar with high anthocyanin. Japan Agricultural Research Quarterly, 46, 321–328.
Round 2
Reviewer 1 Report
Comments and Suggestions for Authors
Thank you for your submission. While the topic is interesting, I do not recommend this paper for publication at this time. My main concerns are:
-
Weak explanation of science
The paper does not clearly explain how NIR relates to leaf tenderness. Since tenderness is a physical property, using NIR requires more support from chemical or texture data. Right now, the interpretation is not strong. -
Insufficient sample validation
The model is based on only 100 samples, from one location. No external validation was used. This makes the results less reliable and difficult to generalize. -
Overstated conclusions
Some parts of the paper claim strong model performance, but the results are not fully supported. Also, important variables like moisture or fiber were not measured directly.
Author Response
Comments 1: The Authors need to explain how NIR measurement correlates to leaf tenderness. This point must be better clarified.
Response 1: Thank you very much for your valuable comments. Near-infrared (NIR) spectroscopy captures resonant (overtone and combination) absorptions related to the chemical composition of leaves during their developmental stages, along with leaf scattering effects. This enables indirect reflection of biochemical composition and tissue structure differences, thereby facilitating the discrimination of tenderness grades in Zijuan tea fresh leaves. NIR mainly corresponds to the overtone and combination vibrations of molecular bonds, which produce characteristic absorption peaks and troughs in the NIR region. In this study, we conducted a detailed analysis of the NIR spectral information of Zijuan tea fresh leaves, as presented in lines 253–263. NIR, as a non-destructive detection technique, has been widely applied in the tea industry. When combined with machine learning or deep learning algorithms, it is primarily used for quantitative prediction of tea constituents and qualitative discrimination of different geographic origins. In this work, we employed NIR technology to achieve both tenderness grade discrimination and key anthocyanin content prediction for Zijuan tea fresh leaves, thereby establishing a solid theoretical foundation for intelligent, source-level tenderness grading in Zijuan tea processing.
Comments 2: The model is based on 100 samples which is adequate, but only from one location. The Authors need to explain this choice, and show how these observations can be generalized.
Response 2: Thank you very much for pointing out that the single-source data used in our model may affect its generalizability. The primary objective of this study was to demonstrate the feasibility of combining NIR spectroscopy with machine vision for discriminating tenderness grades and predicting anthocyanin content in Zijuan tea. Therefore, it was reasonable to first conduct method validation under conditions with high environmental consistency. It is well established that teas produced from the same cultivar grown in different regions exhibit distinct near-infrared spectral profiles. This phenomenon has prompted many researchers to employ NIR spectroscopy for tea origin tracing, in order to combat counterfeit products in the market. In our future work, we plan to expand the study to include multi-location and multi-cultivar tea samples, thereby further verifying the model’s universality and stability. To enhance adaptability, we also plan to introduce transfer learning techniques, enabling cross-domain fine-tuning to achieve effective algorithm migration across different growth environments and tea types. In parallel, we have initiated the construction of a multispectral imaging database, covering spectral variations of tea leaves under different illumination, temperature, and humidity conditions. This will strengthen the model’s ability to adapt to complex real-world application scenarios. These improvements are expected to substantially increase the model’s applicability and provide broader technical support for rapid and non-destructive quality detection of Zijuan tea.
Comments 3: The Authors need to explain why they chose to measure certain variables but not others, such as moisture or texture.
Response 3: We sincerely appreciate your valuable comments regarding our variable selection. Moisture is indeed a critical variable in tea processing, serving as a key criterion for decision-making during withering, rolling, fermentation, and other processing stages. Our research team has previously employed spectroscopic non-destructive testing techniques to determine moisture content in green tea during withering and in black tea fermentation leaves, along with other key quality parameters (Zhu et al., 2019; Dong et al., 2020; An et al., 2020; Luo et al., 2022; Yang et al., 2024; Qi et al., 2024).
Zijuan tea is rich in anthocyanins, which distinguishes it from other tea varieties. In this study, we selected chemical components closely related to the flavor and quality of Zijuan tea—particularly anthocyanins—as the main variables. These compounds not only have a significant impact on the sensory quality of tea but also exhibit notable differences among fresh leaves of Zijuan tea at different tenderness levels. We applied near-infrared spectroscopy to achieve rapid and non-destructive detection of these variables, enabling precise quality monitoring at the source of Zijuan tea processing.
As for variables such as moisture and texture in Zijuan tea, we plan to expand our work in future studies to gain a more comprehensive understanding of the entire processing chain. Fresh Zijuan tea leaves are often processed into different tea types depending on geographic origin and specific processing requirements. When Zijuan tea leaves are processed into various types (such as green tea, black tea, or dark tea) using different processing techniques, the intelligent monitoring of both moisture changes and key anthocyanin content variations throughout the different processing stages becomes a large and complex scientific problem—one that requires a step-by-step approach for researchers to address.
References
Zhu, Hongkai, Fei Liu, Yang Ye, Lin Chen, Jingyuan Liu, Anhui Gui, Jianqiang Zhang, and Chunwang Dong. "Application of Machine Learning Algorithms in Quality Assurance of Fermentation Process of Black Tea-- Based on Electrical Properties." Journal of Food Engineering 263 (2019): 165-72. https://dx.doi.org/10.1016/j.jfoodeng.2019.06.009.
Dong, C., T. An, H. Zhu, J. Wang, B. Hu, Y. Jiang, Y. Yang, and J. Li. "Rapid Sensing of Key Quality Components in Black Tea Fermentation Using Electrical Characteristics Coupled to Variables Selection Algorithms." Sci Rep 10, no. 1 (Jan 31 2020): 1598. https://dx.doi.org/10.1038/s41598-020-58637-9.
An, Ting, Huan Yu, Chongshan Yang, Gaozhen Liang, Jiayou Chen, Zonghua Hu, Bin Hu, and Chunwang Dong. "Black Tea Withering Moisture Detection Method Based on Convolution Neural Network Confidence." Journal of Food Process Engineering 43, no. 7 (2020). https://dx.doi.org/10.1111/jfpe.13428.
Luo, Xuelun, Mostafa Gouda, Anand Babu Perumal, Zhenxiong Huang, Lei Lin, Yu Tang, Alireza Sanaeifar, Yong He, Xiaoli Li, and Chunwang Dong. "Using Surface-Enhanced Raman Spectroscopy Combined with Chemometrics for Black Tea Quality Assessment During Its Fermentation Process." Sensors and Actuators B: Chemical 373 (2022). https://dx.doi.org/10.1016/j.snb.2022.132680.
Yang, C., L. Jiao, C. Dong, X. Wen, P. Lin, D. Duan, G. Li, C. Zhao, X. Fu, and D. Dong. "Long-Range Infrared Absorption Spectroscopy and Fast Mass Spectrometry for Rapid Online Measurements of Volatile Organic Compounds from Black Tea Fermentation." Food Chem 449 (Aug 15 2024): 139211. https://dx.doi.org/10.1016/j.foodchem.2024.139211.
Qi, D., Y. Shi, M. Lu, C. Ma, and C. Dong. "Effect of Withering/Spreading on the Physical and Chemical Properties of Tea: A Review." Compr Rev Food Sci Food Saf 23, no. 5 (Sep 2024): e70010. https://dx.doi.org/10.1111/1541-4337.70010.